# Design and Depth Control of a Buoyancy-Driven Profiling Float

**DOI:** 10.3390/s22072505

**Published:** 2022-03-25

**Authors:** Yulin Bai, Rui Hu, Yuanbo Bi, Chunhu Liu, Zheng Zeng, Lian Lian

**Affiliations:** 1School of Oceanography, Shanghai Jiao Tong University, Shanghai 200000, China; bylsjtu@sjtu.edu.cn (Y.B.); hurui_sjtu@sjtu.edu.cn (R.H.); biyuanbo@sjtu.edu.cn (Y.B.); liuchunhu@sjtu.edu.cn (C.L.); zheng.zeng@sjtu.edu.cn (Z.Z.); 2State Key Laboratory of Ocean Engineering, Shanghai Jiao Tong University, Shanghai 200000, China

**Keywords:** profile float, depth control, thermocline observation

## Abstract

This paper presents the design and fabrication of a profiling float primarily used for thermocline observations and tracking, with an emphasis on depth control performance. The proposed float consists of a frame-type electronic chamber and a variable buoyancy system (VBS) actuator chamber. Components or sensors can be added or removed according to specific requirements. All components were off the shelf, which lowered the cost of the float. In addition, a segment PD control method is introduced. Simulink results showed that there was no need to change any parameter when carrying out tasks at different depths. This method is superior to the traditional PD control and sliding mode control (SMC). In the process of diving, the speed could be well controlled to less than 0.2 m/s. We completed depth determination and control method validation in Qiandao Lake. The final results were consistent with the simulation results, and the maximum depth retention error was less than 0.3 m. Field tests also demonstrated that the prototype float can be used for thermocline observations in the upper layer of seawater or lake water.

## 1. Introduction

Typically, various materials in the upper ocean present significant vertical changes. In the mixed zone, the influence of ocean currents is more pronounced than in deep ocean. Therefore, the upper-middle layer of the ocean is the wealthiest region of physical and biological processes, where most of internal waves, thermocline, and vertical exchange of biological and chemical elements occur [1,2]. Continuous observation of the upper and middle layers of the ocean over long periods is necessary to analyze these phenomena. Among the numerous upper ocean phenomena, the thermocline is worth studying and utilizing [3,4,5]. In the thermocline, seawater goes from top to bottom, with a sudden drop in temperature and a sudden increase in density. This particular property makes the thermocline region a natural sound barrier in the ocean [6]. At the same time, the huge physical differences between the water layers also limit the exchange of water bodies, so the influence of water flow in the vertical direction can be ignored. Therefore, the most important environmental impact is the highly variable density when tracking and maintaining depth in the thermocline. Most ocean-observing missions, including thermoclines, can be performed with floats [7,8,9].

Profiling floats equipped with sensors are a simple solution that allows water column monitoring for a long time and at a low cost. They are controlled generally by changing their buoyancy, which enables them to travel downwards or upwards. They can also be controlled so to reach and maintain a certain depth. They are normally simple systems that have no actuator for horizontal movement. However, floats also have a certain ability to track the horizontal current, which, to a certain extent, can reflect information on the current in deep water [7,10]. According to the different tasks to be performed, a program can be preset to realize actions such as rising, diving, or depth-fixing in seawater. The shape of the float is generally cylindrical [11,12] or spherical [13] with a small cross-sectional area, making the vertical resistance coefficient small in water. Therefore, the float system has low damping and large inertia. The usual buoyancy control system of floats mainly relies on oil pumping or linear actuator movement to change the volume of floats [8,12,13,14,15]. The speed of this process is relatively slow, so it takes a specific time to reach the target drainage volume required by control, which gives the float system a certain time lag. In addition, the sampling rate of some oceanographic sensors on floats is relatively low. Therefore, to ensure the accuracy of data collection in the upper ocean where the environment changes rapidly, it is necessary to control the diving and floating speed of the float. It is also essential to develop a general control method suitable for floats with low damping and large time delay.

The traditional PD method is independent of the float model and is simple and easy to implement [16]. However, this method is based on different parameters to achieve the best control effect in different environments or control tasks [8,14] and avoid overshoot or even failure to reach the control target. At present, the most widely used control methods for the nonlinear float model are based on two approaches. One consists in linearizing a nonlinear system and then achieving control through state feedback [13,14,17] PID and other linear control methods. However, achieving a good control requires a relatively accurate modeling. The other approach is based on fuzzy control [18,19], sliding mode control [20,21], and active disturbance rejection control (ADRC) [22]. Among them, the sliding mode control is highly insensitive to model error, variation of controlled object parameters, and external interference, so it is often used in the motion control of underwater vehicles [20,23]. However, chattering occurs when the state trajectory reaches the sliding mode surface. ADRC has excellent anti-interference ability, but the ADRC controller has many control parameters and cannot be calibrated fast and effectively, thus requiring long time and high costs in practical applications.

In this study, a small depth linear actuator piston-type float was built to continuously observe and track the thermocline. The float was equipped with a temperature-depth sensor, a 4G or Beidou communication system, a GPS positioning system, and a variable buoyancy system (VBS). All modules necessary to build the float were off-the-shelf and low-cost, which provides the possibility of mass production and deployment. In addition, the electronic bin and the buoyancy bin of the float adopted a detachable installation method, which was small in size and light in weight, easy to disassemble and assemble, and easy to transport. Furthermore, in order to fulfill the requirements of depth keeping and speed control, a segmented PD control method is proposed in this paper. This method enabled the float to reach the target depth at the desired speed and remain at the target depth. In addition, to reach different depths in the same experimental environment, the method does not need to change the parameters, which greatly limits time and cost of the experiment. The float was tested at depths between 5 and 60 m in Qiandao Lake, confirming its speed tracking capability and depth retention capability. Finally, the distribution characteristics of the thermocline in the Qiandao Lake were determined by using the temperature data of the float during the depth control process at three different positions.

## 2. System Design

The overall design goal was to create a system that controlled buoyancy through a linear actuator and could accurately maintain depth and control speed through a low-cost depth sensor. Each part of the float system was modular in design, allowing different observation instruments and communication methods to be selected according to different observation scenarios.

The hardware system used was mainly composed of VBS, communication system, positioning system, battery system, electric control system, and sensor unit. The details and distribution of the specific parts can be seen in Figure 1a,b, showing the SolidWorks perspective view on the back of the float (a), and the physical picture of the float on the front (after removing the transparent cylindrical acrylic tube (b)).

### 2.1. Variable Buoyancy System

The VBS included the linear actuator, a driver board, a piston, and an external metal tube. When the linear actuator was pushed outward, the volume of the float became larger, and the static buoyancy increased, causing the float to float up. The volume could vary up to 4.81 × 10^−4^ m^3^, accounting for 5.7% of the maximum volume. The maximum volume change rate was 1.03 × 10^−5^ m^3^/s.

### 2.2. Communication System and Positioning System

The communication system consisted of two parts, which communicated with the main control board through a Universal Asynchronous Receiver/Transmitter. The connection between the float and the computer through the duplex Bluetooth module was realized before the float was formally applied. The acrylic material used in the electronic warehouse did not block the Bluetooth signal, so data could be transmitted or read without dismantling the warehouse. The program’s scheduled task could also be burned through this module. The second part was responsible for sending back the environmental data collected by the float after it rose to the surface. The float system could choose different communication modules according to different application scenarios. For example, 4G communication modules have higher single transmission capacity and higher transmission frequency but can only work in areas covered by 4G signals. Meanwhile, the satellite communication adopted by Beidou short message requires one minute between two data transmissions and a small amount of single transmission. However, Beidou short message communication is not restricted by region, so as long as it can receive satellite signals, the communication needs can be met. Therefore, the 4G communication module and GPS positioning system can be used for float communication in inland lakes or adjacent seas, whereas Beidou short message communication and positioning module can be used for communication in the ocean.

### 2.3. Battery System

The battery system consisted of a 14.8 V, 10,000-mAh, 25C, 4s LiPo battery and a galvanometer. The LiPo battery was used to power electronics and VBS. The galvanometer was used to measure the remaining battery power and the power of the system operation. When the battery voltage was too low, the float linear actuator was moved to the outermost position, so that the float could move up to the surface of the water for recovery.

### 2.4. Sensor Unit

Sensors can be added to the float according to observation requirements. The most basic sensor is a low-cost temperature depth sensor with a depth range of 0–300 m, a temperature range of −20–85 °C, a depth accuracy of 0.01 m. This sensor is used for temperature and depth measurement and can provide the approximate value of speed by subtracting two adjacent depth data. However, the accuracy of velocity obtained by doing so is relatively low. Reducing the sampling frequency can effectively increase the speed accuracy, but it also reduces the control rate. Therefore, when designing the controller in the later segment PD, if speed is needed for the float control, the sampling frequency of the sensor should be set to 1.43 Hz; otherwise, it should be set to 2.78 Hz. Adding a speed sensor can undoubtedly improve the control accuracy, but it will also correspondingly increase cost and power consumption. In this paper, the precise control of float depth was achieved only by a low-resolution temperature and depth sensor through the variable target segment PD control method.

## 3. Depth Control System

When the float performs tasks in water, it will automatically change the force in the vertical direction, resulting in floating and diving. In the motion process, it will be affected by the flow, the buoyancy change caused by the change of seawater density, and the deformation of the electron chamber. The effect of the current on the float is mainly reflected in the horizontal direction, which makes the float have a specific ability to follow the current, and its influence on the vertical direction can be ignored. Therefore, in the study of depth control, the impact of water flow was ignored, and the change of seawater density and the deformation of the electronic warehouse were taken into consideration.

### 3.1. Motion Modeling

In the process of modeling, vertical down is defined as the positive direction of the coordinate. The motion equation of the float can be established by Equation (1) through the vertical gravity, buoyancy, and resistance, as shown in Figure 2.
(1)$$M\ddot{h}=G-F_{b}-F_{r}$$
where M is the mass of the float, h is the depth of the water where the float is located, G is gravity, G=Mg, g = 9.8 m/s^2^, $F_{b}$ is buoyancy, and $F_{r}$ is resistance.

The resistance can be calculated by Equation (2); by replacing $v^{2}$ with $v\cdot \left|v\right|$, the direction of resistance is obtained.
(2)$$F_{r}=\frac{1}{2}C_{r}v\left|v\right|\rho A$$
where A Is the cross-sectional area of the float, v is the speed of the float in the water, and $C_{r}$ is the drag coefficient, which is related to the shape and speed of the float [13]. The expected speed of the float in the process of diving was not more than 0.2 m/s. Through ANSYS, the resistance $F_{r}$ of the float under the condition of 0.2 m/s can be calculated. The drag coefficient can be obtained by further transforming (2) into (3). The final $C_{r}$ value was 0.66.
(3)$$C_{r}=\frac{2F_{r}}{\rho Av^{2}}$$

The density of seawater is influenced by temperature, salinity, and pressure. In the upper layer of water, the change of density is more drastic, so the influence caused by the change of seawater density could not be ignored in the simulation. Equation (4) indicates the density change of the upper layer of seawater fitted with a group of Argo floats.
(4)ρ=0.02073h+1023

The buoyancy of a float (5) is affected by the density of the liquid at its location and the volume of water pumped out. The volume change of the float has two components. One is the active volume change caused by the float pushing the linear actuator, and the other is the passive volume change caused by the deformation of the body with the change of depth. In Equation (6) $V_{d}$ consists of the initial volume of the float $V_{0}$, the volume of active change $\Delta V_{r}$ and the volume of deformation $\Delta V_{P}$.
(5)$$F_{b}=\rho V_{d}g$$
(6)$$V_{d}=V_{0}+\Delta V_{r}+\Delta V_{P}$$

The deformation of the float depends on the material selected and the pressure applied. For an acrylic cylinder, the compression deformation can be calculated by Equation (7)
(7)$$\Delta V_{P}=2\pi R_{o}\cdot \Delta R_{o}\cdot L+\pi R_{o}^{2}\Delta L$$
where $R_{o}$ is the external radius of the cylinder, and L is the length of the cylinder, and its variation can be calculated by Equations (8) and (9) [24].
(8)$$\Delta R_{o}=\frac{R_{o}\cdot P}{E\cdot \left(R_{i}^{2}-R_{o}^{2}\right)}\times \left[\left(1-2\mu \right)R_{i}^{2}+\left(1+\mu \right)R_{o}^{2}\right]$$
(9)$$\Delta L=\frac{R_{o}{}^{2}\cdot P}{E\cdot \left(R_{i}^{2}-R_{o}^{2}\right)}\times \left[\left(1-2\mu \right)L\right]$$
where P is the pressure, E is the elastic modulus of the material, $R_{i}$ is the internal radius of the cylinder, and μ is the Poisson’s ratio.

The specific values and units of each parameter are reported in Table 1.

### 3.2. Control Strategy

When designing the controller, the following four aspects should be considered. First, the cross-sectional area of the float is very small, and the characteristics of seawater make the float system have low damping and large inertia in seawater. Second, due to the limited putter speed, the system has an inevitable time delay. Third, in order to quickly deploy floats, the controller should be capable of adapting to different depths without changing the parameters. Finally, the sampling frequency of most ocean observation sensors is relatively low, and seawater parameters vary dramatically in the thermocline region. Therefore, to ensure the quality of the data collected by floats, it is generally required that the diving or floating speed of floats should not be greater than 0.2 m/s.

The traditional PID control system is the most widely used control method in the industry. The differential part of a traditional PID control can solve the system characteristics of time delay. In contrast, the integral part will increase the time delay and overshoot in the position control process. Therefore, the comparison control algorithm and the control algorithm proposed in this paper are based on PD control. For a system with large inertia and time delay like a float, the differential term overcomes this difficulty to a certain extent and has good robustness. However, it may be necessary to reset the parameters when the target depth changes, which will be an enormous waste of time. In addition, when tracking the target depth, the speed of the float cannot be effectively controlled. Therefore, when the targeted depth cannot be reached, and the floating speed will be too fast, which will affect the quality of data acquisition.

In essence, the direct output result by the controller is expressed as changing the static buoyancy of the float, that is, the acceleration of the float. Therefore, inertia will be greater when the depth error is used as the PD controller input than when the velocity error is used as the PD controller input. In order to reduce the inertia, the velocity error can be considered as the input of the PD controller. The actual velocity can be obtained by the difference approximation of the depth sensor, and the target velocity can be obtained by designing a curve of the target velocity changing with the depth error through the function relationship. This curve should be an odd function curve that takes the error between the target depth and the actual depth as the x-axis and the target velocity as the y-axis and passes through the origin. When the depth error is greater than zero, the target speed must be greater than zero. There are various functions that satisfy this condition. In this study, Equation (10) was adopted.
(10)$$v_{d}=\frac{e_{d}}{a\left(b+\left|e_{d}\right|\right)}$$
where $e_{d}$ is the error between the target depth and the current depth, and a and b are two constant parameters. The value of a can be determined by the desired speed when the depth is far from the target. The value of b can be determined by the expected velocity at a given position. The values of the parameters a and b in the subsequent simulation and experiment were 5 and 5.67, respectively.

Theoretically, depth control can be achieved by the above method. However, as the float speed determination depends on the depth sensor and is limited by the program’s cycle time and the depth sensor’s accuracy, the accuracy of the speed was about 0.01428 m/s. Floats can no longer accurately track the target speed with less than twice the speed accuracy. At this point, the float was about 0.8 m from the target depth, and the speed of the float was sufficiently low. So, depth PD control was adopted within the range of 0.8 m from the target depth. Due to the better robustness of the PID system, the PD parameters of two stages at different depths could achieve better depth control accuracy and speed tracking without adjustment in the same or similar sea area.

In general, for segment PD control, in the initial stage of diving, the speed PD control with low inertia enables the float to dive at the expected speed and ensure that the speed is small enough when it is close to the target depth. Depth PD control was used in the area close to the target depth to avoid insufficient accuracy of differential velocity. The flow chart of the whole control process is shown in Figure 3. When the current position of the float was more than 0.8 m away from the target depth ($e_{d}$ > 0.8), the float was controlled according to the red line in Figure 3. First, the depth error $e_{d}$ was used to calculate the target velocity $v_{d}$ through Equation (10), and then the difference between target speed $v_{d}$ and actual speed $v_{r}$ was used as the input of PD to control the float. When $e_{d}$ < 0.8, the float was controlled according to the blue line in Figure 3. Then, the control strategy was equivalent to the traditional PD control.

## 4. Results and Analysis

In order to verify the feasibility of the control method mentioned above, a dynamic model of the float and a linear actuator system model were constructed by Simulink, considering the change of seawater density and the deformation of the electronic chamber. The traditional PD method for depth error, the segment PD method proposed in this paper, and SMC were used to compare the depth control of the float at 20 m, 50 m, and 80 m, respectively. Subsequently, field tests were carried out at three locations in Qiandao Lake, Zhejiang Province, China. Continuous depth control and temperature data ranging from 5 m to 60 m were obtained using the segment PD control proposed in this paper. Finally, the application of the float in thermocline measurement was demonstrated.

### 4.1. Simulation Results and Analysis

The proposed control method was compared in this simulation with PD control and SMC, which are currently most the commonly used methods in industrial processes. In order to simulate the actual process, the following comparison was adopted.

First, the target depth of depth control was set to 20 m, which is relatively shallow. The parameters of the three methods were adjusted to ensure that the final depth control performance was good. In addition, it was necessary to ensure that the speed of the float during the diving process did not exceed 0.2 m/s. In this paper, the parameters of traditional PD control were set as $K_{p}$ = 1 and $K_{d}$ = 70. In the speed control process of segment PD control, the parameters were set as $K_{p}$ = 1, $K_{d}$= 2, and in the depth control process, as $K_{p}$ = 1, $K_{d}$= 150. For SMC, the sliding surface was set as s=k⋅x1+x2, where k is the sliding surface coefficient, which was taken as 0.1 in this paper, x1 is the depth, and x2 is the speed. The approach rate is $\dot{s}=-\varepsilon \cdot \mathrm{sgn}\left(s\right)$, where ε is a constant greater than zero, which was 0.2 in this paper.

Then, without changing the parameters, simulations with target depths of 50 m and 80 m were carried out. The final simulation results using the three depth control methods are shown in Figure 4. The three graphs from left to right in Figure 5 show the speed curves of the three control algorithms during the depth control process t 20 m, 50 m, and 80 m, respectively.

As can be seen in Figure 4 and Figure 5, when the depth target was 20 m, since the parameters of each method were adjusted to the optimum, both segment PD control and SMC could dive at a speed not exceeding 0.2 m/s and successfully determine the depth. The SMC stabilized faster. For the traditional PD method, in order to control the diving speed and reduce the depth oscillation of the float in the fixed depth stage, the coefficient of the differential term was set to be very large. This also caused the traditional PD method to converge slowly when approaching the target depth. When the target depth was 50 m, the results of segment PD control and SMC control were similar to those obtained at 20 m. While the traditional PD control did not change the parameters as the target depth increased, the proportional term played a leading role in the first half of the dive, resulting in a speed significantly greater than 0.2 m/s in 15–100 s. This is not acceptable for some ocean observation sensors. When the target depth was 80 m, the convergence time of segment PD control was significantly smaller than that of SMC. This is because, when the target depth was small for SMC, the initial state was closer to the sliding mode surface, so a small approach rate could meet the requirements of quickly reaching the sliding mode surface and converging. However, as the target depth increased, the initial state was farther and farther away from the sliding surface, and the convergence time increased significantly with the increase of the target depth, without changing the approach rate. However, the traditional PD method showed the shortest final convergence time due to the fast speed in the initial dive.

In addition, since the float system was damped low and the linear actuator had a certain time delay, the float was not stable at the target depth but oscillated up and down, as shown in the enlarged areas in the lower parts of Figure 4 and Figure 5. The control strategy of SMC was similar to the switch structure, vibrating back and forth at the sliding surface, so the final vibration amplitude was the smallest. However, this vibration had a significant impact on the actuator. For the traditional PD control, the larger the differential term, the smaller the amplitude of the float vibration, but the parameter value of the differential term will also affect the convergence time. In intermediate conditions, the float ends up oscillating the most. The segment PD control was equivalent to the traditional PD control when the depth is less than 0.8 m, but the differential term coefficient could be appropriately increased. So, the oscillation range was between those of the other two methods.

The segment PD control ensured that the float could always dive at a speed close to 0.2 m/s during the initial stage of the dive and achieved the purpose of speed control. As regards the speed control ability, it can be seen in Figure 6 that the speed tracking was accurate before 380 s. After that, the velocity error was used as input, and the oscillation of the float at 50 m can be seen more clearly.

Therefore, it can be concluded that the segment PD method can realize the speed control in the process of diving, and it is not necessary to change the parameters in similar sea conditions or for different target depths. Compared with the traditional PD control, it saves time and allows speed control. Without changing the parameters, when the target depth is larger, the convergence time is shorter compared to SMC.

### 4.2. Field Experiments

In order to verify the feasibility of the segment PD control, a depth control test t different depths was carried out in three different areas of Qiandao Lake, Zhejiang Province, China, in July. The depth holding time was set to 1 min at the depths of 5 m and 10 m, and 1.5 min at other depths. After the depth setting was completed, the linear actuator was pushed to the outermost side to make the float move to the surface of the lake. In the speed control process of segment PD control, the parameters were set as $K_{p}$ = 1, $K_{d}$ = 1.7, and in the depth control process, they were set as $K_{p}$ = 1, $K_{d}$ = 13.

The final depth retention results at different depths are shown in Figure 7. The process before the hollow red dot is the speed control stage, and the subsequent process is the depth holding stage with the depth error as the PD input. It can be seen that the actual depth retention effect was consistent with the simulation results. We found that 80% of the final depth retention error was within 0.2 m, and the maximum error did not exceed 0.3 m. One set of parameters was suitable for the depth retention at each depth, which reflects the good control of the segmented PD control method for a low-damping and large time-delay system. In addition, the magnified image in the upper left corner shows a simulated throwing of a float. After being thrown into the water, the float first reached a depth of 2 m due to inertial depth, then floated for a short time, and finally dived and maintained a depth of about 10 m. It was found that the throwing layout did not affect the final depth control effect, even if the target depth was very shallow. Figure 8 shows the tracking speed of the float during the retention process at a depth of 60 m. The blue line is the target speed to be tracked. The faint yellow line is the actual speed data that were not processed. The green line is the actual speed curve of the float after passing through the FIR low-pass filter. The red area is the range where the point of the actual float speed fell. It can be seen that the float had a good speed tracking effect and met the requirements.

In addition, by comparing the temperature data of the ascending and descending portions, it was found that the temperature during diving at the same water depth was generally higher than that during ascending. The reason may be as follows: the speed was always kept within 0.2 m/s when diving, the linear actuator was directly pushed to the outermost side when floating, and the floating speed of the float was very high, which affected the temperature sensor, with a low sampling rate. This shows that it is necessary to control the speed during the ascent and the descent of the float.

### 4.3. Thermocline Determination

The thermocline is one of the most common phenomena in the upper layer of water. Nowadays, there are two most commonly used methods for determining the thermocline: one is based on the average temperature gradient [4], and the other is based on the peak temperature gradient [3,5].

In the depth control of the three locations in Figure 9, the maximum depth was 60 m (red marker), followed by 40 m (green marker) and 10 m (yellow marker). These three depths are very close to the highest local water depth. The temperature data during the diving process of the float were processed through the average temperature gradient method, and the thermocline conditions of the three places were obtained. The final measurement results are shown in Figure 9. The color bar represents the variation of temperature with depth. The thermocline at a water depth of 40 m accounted for the largest proportion of the total depth, as shown in Table 2. It was found that the range of the thermocline at water depths of 40 m and 60 m was very similar, and the temperature of the thermocline changed dramatically in the upper and lower portions, while the temperature in the middle portions changed relatively little. In the area over 40 m, the water temperature changed slowly. In the area with water depth of 10 m or so, the temperature changes in thermocline were more evident and uniform.

## 5. Conclusions

This paper introduced a linear actuator piston float for continuously observing and tracking the thermocline. The buoyancy system and electronic system of the float adopted a modular design scheme, and other sensors could be installed according to specific needs. This design has the advantages of easy assembly, easy integration, portability, and low cost. In addition, to meet the observation requirements, a segmented PD control method was proposed. The simulation results showed that the control algorithm was superior to that of the traditional PD control. When the target depth was high, the segment PD could converge faster than the SMC. Finally, the algorithm proposed in this paper could achieve an accurate control of speed and depth. The field experiments were consistent with the simulation and proved that this float could be used for thermocline observations.

## Figures and Tables

**Figure 1 sensors-22-02505-f001:**
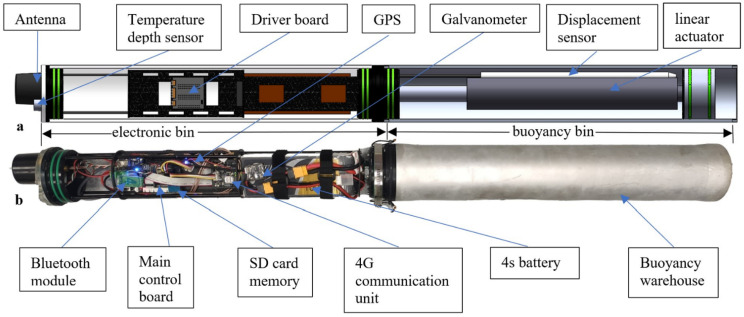
Hardware composition and layout of the float.

**Figure 2 sensors-22-02505-f002:**
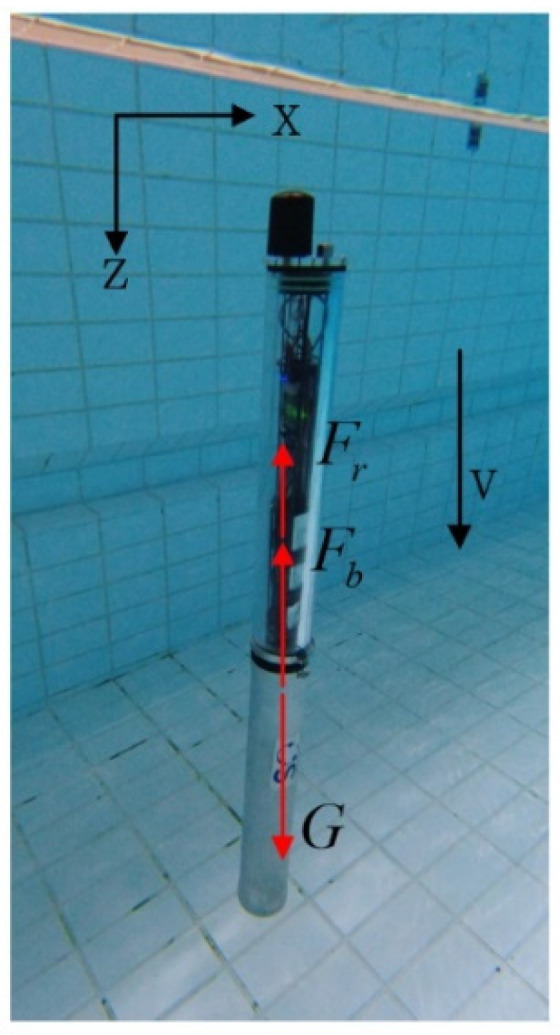
Main forces acting on a moving float.

**Figure 3 sensors-22-02505-f003:**
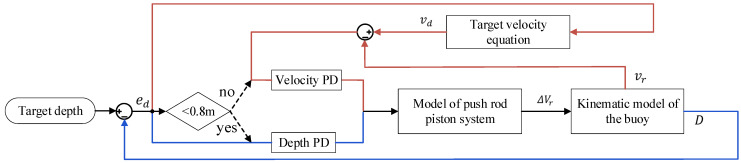
Schematic diagram of segment PD control.

**Figure 4 sensors-22-02505-f004:**
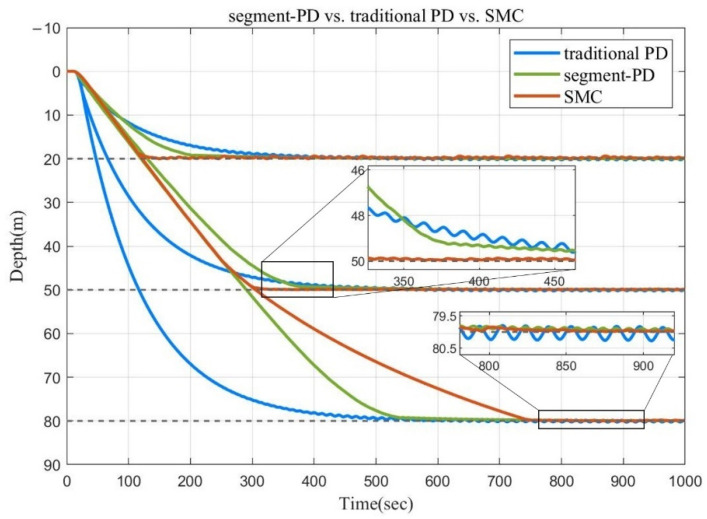
Control performance of the three depth control methods at the target depths of 20 m, 50 m, and 80 m.

**Figure 5 sensors-22-02505-f005:**
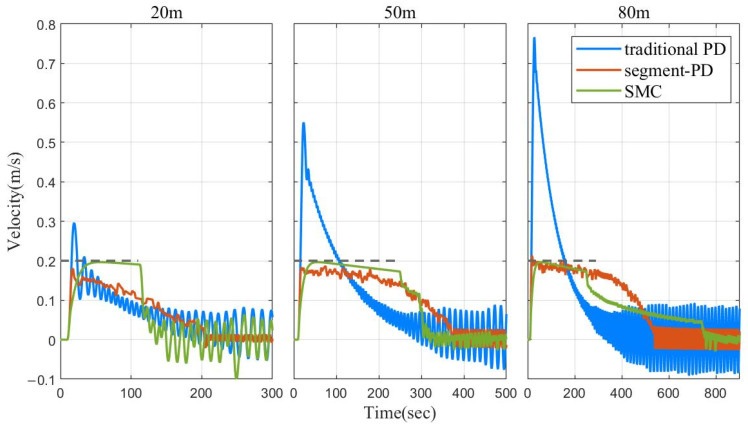
Speed comparison of the three depth control methods in the processes of float diving and depth control.

**Figure 6 sensors-22-02505-f006:**
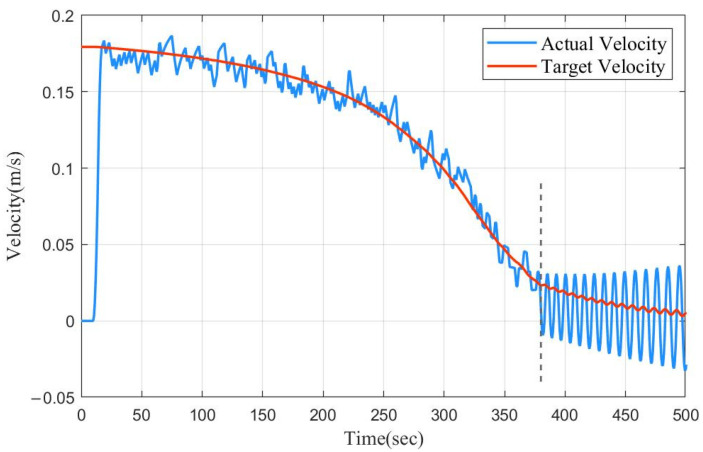
Speed tracking in depth control by segment PD control (target depth was 50 m).

**Figure 7 sensors-22-02505-f007:**
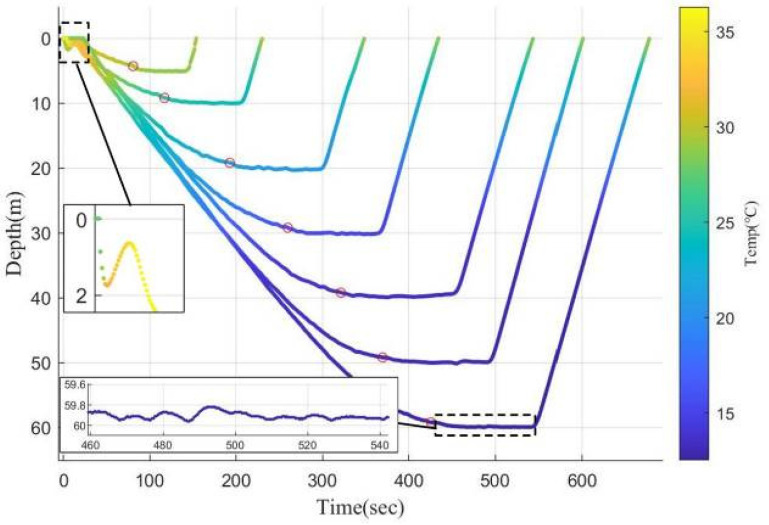
Using the segment PD control method to maintain the float at different depths in Qiandao Lake.

**Figure 8 sensors-22-02505-f008:**
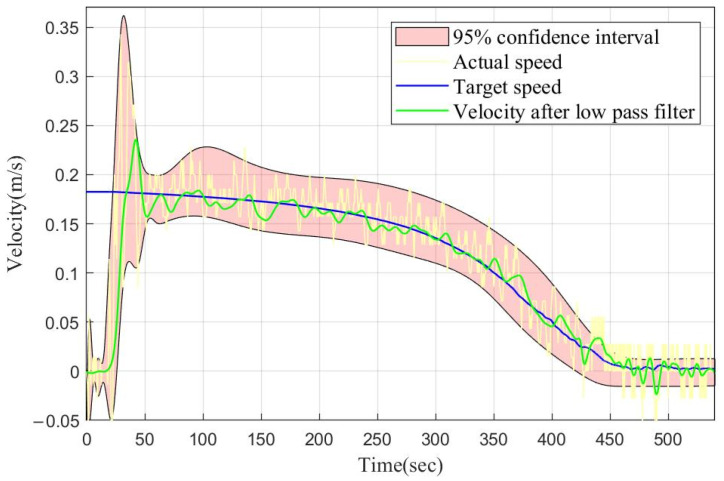
Field test of velocity tracking during the float descent to a depth of 60 m.

**Figure 9 sensors-22-02505-f009:**
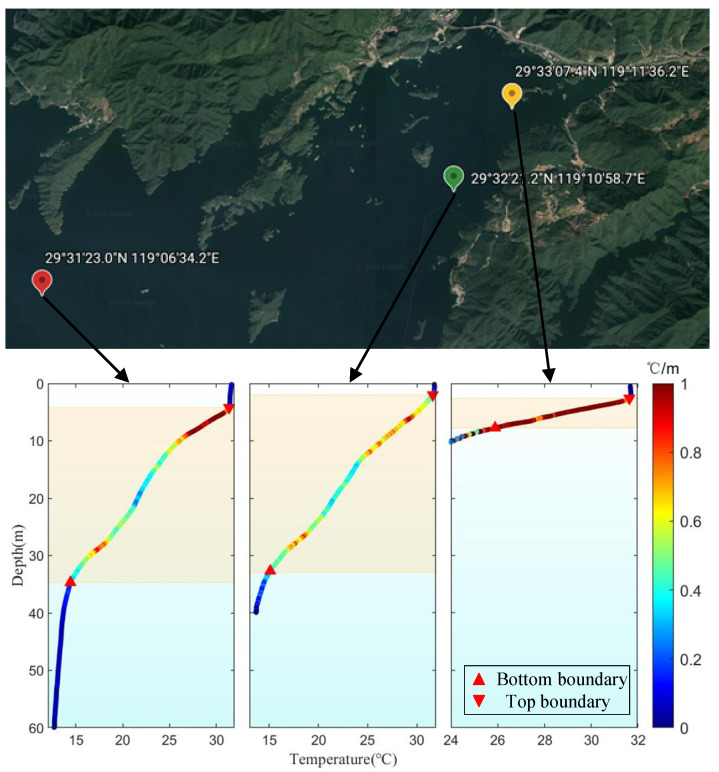
Thermocline measurements at three different locations in Qiandao Island Lake using floats.

**Table 1 sensors-22-02505-t001:** Float dynamic parameters.

Parameter	Description	Value	Units
M	Float mass	6.29	kg
$C_{r}$	Coefficient of drag	0.66	-
A	Cross sectional area	0.00785	m^3^
L	Electron bin length	51.5	cm
$R_{o}$	Outer diameter of electronic bin	0.05	m
$R_{i}$	Internal diameter of the electron bin	0.04	m
$V_{0}$	Initial float volume	0.00839	m^3^
μ	Poisson ratio	0.32	-
E	Elasticity modulus	3.16	GPa

**Table 2 sensors-22-02505-t002:** Thermocline properties.

Measured Depth	Top Boundary	Bottom Boundary	Thermocline Thickness Ratio
60 m	4.2 m	34.68 m	50.8%
40 m	1.94 m	32.96 m	77.55%
10 m	2.55 m	7.7 m	51.5%

## Data Availability

The raw/processed data required to reproduce these findings cannot be shared at this time, as the data are also part of an ongoing study.

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
