# Peer review of "Design and Depth Control of a Buoyancy-Driven Profiling Float"

_sensors, 2022, doi:10.3390/s22072505_

Round 1

Reviewer 1 Report

The authors have proposed a low-cost buoyancy-driven float with a linear actuator variable buoyancy system. To accurately adjust the depth of the float system in upper layer water, they have designed a segmented PD controller to tackle the system's nonlinear features and the water density changes. The proposed control method was verified efficiently by numerical simulations and the field experiments in Qiandao Lake and showed superior control performance of speed tracking. In general, this paper was overall well written and sufficient details were provided on the controller design. However, I have a few minor comments that can be addressed before the publication: 

  1. According to Section 2.4, the float speed is approximated by subtracting two adjacent depth data. The author said, “Reducing the sampling frequency can effectively increase the speed-accuracy but also reduce the control rate”(lines 131-132). But why does reducing the sampling frequency can increase the speed accuracy? Generally, the faster and more data the sensor sampling, the more accurate the result.

  1. I found that the drag coefficient of float Cr is given as 0.033 in Table 1. But the paper also gave the formula (3), which is related to float speed and drag force, to calculate it. So, how the value 0.033 was obtained or estimated? The drag coefficient can be estimated by using CFD or with tow-tank experiments. If the author referred to some related articles to choose the value, these articles should be given in the reference section.

  1. Some writing mistakes should be fixed. For example, the value “a” in line 211 should be set as italic; the speed unit in line 217 “01428.m/s2” should be modified as “0.01428 m/s2”.

  1. Figure 4 shows a significant overshoot of the depth control curve of the traditional PD at 70 m, which did not appear at 30 m and 50 m. It seems the overshoot increases with the target depth of float. This requires an explanation in the paper.

  1. The author should give the parameters applied in the simulation, including the control parameter of the two PD controllers and the constants a and b in the function (10). In addition, Figure 4 verifies that the proposed controller can control the float to target depth without overshooting. But the proposed controller does not seem to have an advantage in fastness compared to the traditional PD.

  1. Today, there are many depth control methods for underwater vehicles, such as linear quadratic regulator (LQR), sliding mode control (SMC), active disturbance rejection control (ADRC) and fuzzy control, etc. Among them, especially SMC and ADRC controllers have excellent anti-interference ability to underwater disturbances. However, this paper only compares with the traditional PD method, which I think is insufficient. Besides, the anti-interference ability of the proposed controller is also crucial since ocean currents may interfere with the float hovering, but this paper did not concern.

  1. I found that the red Target Velocity curve in Figure 5 also has a tiny oscillation after 380 seconds. The target velocity was calculated by the function (10), so why does its curve oscillate with time?

  1. There is a steady-state error of about 0.1 m in the magnified image in the lower-left corner of Figure 6. Is it caused by the insufficient accuracy of the depth sensor? An explanation should be given in the paper.

Author Response

Response to Reviewer 1 Comments

Please refer to the attachment for the revised version.

Point 1: According to Section 2.4, the float speed is approximated by subtracting two adjacent depth data. The author said, “Reducing the sampling frequency can effectively increase the speed-accuracy but also reduce the control rate”(lines 131-132). But why does reducing the sampling frequency can increase the speed accuracy? Generally, the faster and more data the sensor sampling, the more accurate the result.

Response 1: Usually, if the speed of the float is directly obtained by the speed sensor, the faster the sampling frequency, the more continuous speed data can be obtained. However, because the method to obtain the speed of the float in this article is the difference of depth, the accuracy of the depth sensor is 0.01m, and when the depth sampling frequency is 1.43 Hz, it can be roughly considered that the resolution of the obtained speed is roughly 0.01/(1/1.43) = 0.0143, and when the sampling frequency is 2.78Hz, this value becomes 0.01/(1/2.78) = 0.0278. It can be seen that the longer the sampling period, the higher the speed resolution. This result is determined by the nature of the differential and the accuracy of the depth sensor.

Point 2: I found that the drag coefficient of float Cr is given as 0.033 in Table 1. But the paper also gave the formula (3), which is related to float speed and drag force, to calculate it. So, how the value 0.033 was obtained or estimated? The drag coefficient can be estimated by using CFD or with tow-tank experiments. If the author referred to some related articles to choose the value, these articles should be given in the reference section.

Response 2: The drag coefficient values in the previous table are based on the data in (Mu, W.; Zou, Z.; Liu, G.; Yang, Y.; Shi, L., Depth Control Method of Profiling Float Based on an Improved Double PD Controller. IEEE Access 2019, 7, 43258-43268.). In order to ensure the accuracy, according to the suggestion, we calculated the resistance of the float under the working condition of 0.2m/s in ANSYS, and then obtained the latest resistance coefficient according to formula (3).

Point 3: Some writing mistakes should be fixed. For example, the value “a” in line 211 should be set as italic; the speed unit in line 217 “01428.m/s2” should be modified as “0.01428 m/s2”.

Response 3: Thanks for pointing out the mistakes. Related errors have been corrected and the rest of the content has been checked again.

Point 4: Figure 4 shows a significant overshoot of the depth control curve of the traditional PD at 70 m, which did not appear at 30 m and 50 m. It seems the overshoot increases with the target depth of float. This requires an explanation in the paper.

Response 4: Since the previous PD parameter tuning has little comparison significance with the method proposed in this paper, the parameters of the differential term are modified to make it more comparative.

In the previous simulation, overshoot appeared with the increase of depth. This is because when the target depth is deep, the P in the PD control in the early stage of the float dive plays a leading role, so that the piston is in the position of minimum buoyancy during the dive. When the target depth is approaching, the differential term of the PD control comes into play, but due to the limitation of the linear actuator, the piston can not reach the target position quickly, resulting in overshoot. When the depth is shallow, the time that P dominates is relatively short, and the D link works when the piston has not reached the minimum position, so that the piston can reach the target position in a relatively short time without overshooting.

Point 5: The author should give the parameters applied in the simulation, including the control parameter of the two PD controllers and the constants a and b in the function (10). In addition, Figure 4 verifies that the proposed controller can control the float to target depth without overshooting. But the proposed controller does not seem to have an advantage in fastness compared to the traditional PD.

Response 5: The relevant parameters have been added in the paper.

One of the reasons for adopting the segmental PD control strategy is that this method can maintain the speed of the float less than 0.2m/s when diving, which is very important for the sampling quality of various sensors mounted on the float in the future. The traditional PID is too fast during the dive of the float, so the time required to reach the target depth is relatively shorter. In addition, although the time for the traditional PD to reach the target depth is relatively short, the position of the piston is excessively reduced at the position of low water pressure, and the total displacement of the piston is correspondingly increased at the position of high water pressure. (It is equivalent to not making full use of the force of water pressure to reduce the volume of the float, but also overly resisting the water pressure work) This will increase a lot of power consumption.

Point 6: Today, there are many depth control methods for underwater vehicles, such as linear quadratic regulator (LQR), sliding mode control (SMC), (ADRC) and fuzzy control, etc. Among them, especially SMC and ADRC controllers have excellent anti-interference ability to underwater disturbances. However, this paper only compares with the traditional PD method, which I think is insufficient. Besides, the anti-interference ability of the proposed controller is also crucial since ocean currents may interfere with the float hovering, but this paper did not concern.

Response 6: SMC is added to the comparative control algorithm.

The main task of the float is to track the thermocline in the water body. The effect of flow is not considered because the vertical water is stable in the thermocline and the effect of flow can be ignored.

Point 7: I found that the red Target Velocity curve in Figure 5 also has a tiny oscillation after 380 seconds. The target velocity was calculated by the function (10), so why does its curve oscillate with time?

Response 7: Because at the target depth, the float will still have a small amplitude of oscillation. The reason of this oscillation is explained in the paper. The target speed is actually a function of the depth error. Due to the oscillation of the float, the target speed also oscillates after 380s.

Point 8: There is a steady-state error of about 0.1 m in the magnified image in the lower-left corner of Figure 6. Is it caused by the insufficient accuracy of the depth sensor? An explanation should be given in the paper.

Response 8: This phenomenon is not caused by the depth sensor, but partly because the time delay of the system is already very large, so the integral term that may produce the hysteresis effect is not introduced in this paper. At the same time, in order to reduce the chattering after the fixed depth, the differential term in the PD is set relatively large. The result of the above two aspects is that the steady-state error cannot be eliminated very well. But the error will not change with the change of target depth and is within the acceptable range.

Reviewer 2 Report

The article under review is devoted to the development of a depth controller for a buoyancy-driven profiling float. The studies were carried out by the method of mathematical modeling using the Matlab Simulink program, as well as field experiments at Qiandao Lake, Zhejiang Province, China. Experimental studies have confirmed the adequacy of the proposed solutions.

In general, this is an interesting work in which the actual problem is considered. The results of the presented studies may be useful to specialists in the field of underwater vehicle control systems for monitoring the upper layers of lakes or the ocean.

However, I would like to point out to the authors a few shortcomings in writing the text, and also ask you for a few clarifications:

  1. In the Introduction and review, the current state of the research issue is not fully disclosed. In the References are few sources and at the same time 12 out of 17 were published earlier than 2017 (more than 5 years ago).
  2. It is not clear why the authors assumed the possibility of not taking into account the impact of water flow in the study of depth control?
  3. Where did the coefficient ½ in equation (2) come from? Why, when passing from equation (2) to equation (3), the factor v∙|v| was replaced by ?
  4. The paper does not sufficiently describe the principle of control strategy proposed by the authors. The schematic diagram of segment-PD control shown in Figure 3 requires detailing.
  5. The choice of PD regulation, and not P, not PI and not PID, requires an explanation.
  6. Figure 5 shows the overshoot when the depth control of the float is at 70 m, while there is no overshoot when the depth control of the float is at 30 m and 50 m. This requires further explanation.

Author Response

Response to Reviewer 2 Comments

Please refer to the attachment for the revised version.

Point 1: In the Introduction and review, the current state of the research issue is not fully disclosed. In the References are few sources and at the same time 12 out of 17 were published earlier than 2017 (more than 5 years ago).

Response 1: As recommended, we have supplemented the application background, working principle, system characteristics and control strategy of the float in the introduction. And added the latest related literature.

Point 2: It is not clear why the authors assumed the possibility of not taking into account the impact of water flow in the study of depth control?

Response 2: The future application of the float is to continuously observe and track the thermocline. Because the density gradient of the thermocline changes greatly, the physical properties between the layers in the vertical direction are relatively stable, and the current effect is weak. Therefore, in this paper the influence of ocean currents on floats is not considered.

Point 3: Where did the coefficient ½ in equation (2) come from? Why, when passing from equation (2) to equation (3), the factor v∙|v| was replaced by v2?

Response 3: The coefficient ½ is derived from the literature (e. a. D 'asaro, 'Performance ofautonomous Lagrangian floats,' j. atmos.ocean. Technol., Vol. 20, No. 6, pp. 896 -- 911, 2003. & L. Barker, ' 'Closed-loop buoy Control for a Coastal Profiling Float,' MBARI Intern Rep., Moss Landing, CA, USA, 2014, No. 8, Pp. 1-15.).

The absolute value sign is added to formula (2) to indicate the direction of resistance. There are no negative values for the drag coefficient itself. When calculating the resistance coefficient, the resistance direction selected in formula (3) is the default positive direction, which also conforms to the physical meaning of the resistance coefficient.

Point 4: The paper does not sufficiently describe the principle of control strategy proposed by the authors. The schematic diagram of segment-PD control shown in Figure 3 requires detailing.

Response 4: Figure 3 is optimized and explained in additional lines 245-251.

Point 5: The choice of PD regulation, and not P, not PI and not PID, requires an explanation.

Response 5: The reasons have now been explained in the article (lines 204-208).

Point 6: Figure 5 shows the overshoot when the depth control of the float is at 70 m, while there is no overshoot when the depth control of the float is at 30 m and 50 m. This requires further explanation.

Response 6: Since the previous PD parameter tuning has little comparison significance with the method proposed in this paper, the parameters of the differential term are modified to make it more comparative.

In the previous simulation, overshoot appeared with the increase of depth. This is because when the target depth is deep, the P in the PD control in the early stage of the float dive plays a leading role, so that the piston is in the position of minimum buoyancy during the dive. When the target depth is approaching, the differential term of the PD control comes into play, but due to the limitation of the linear actuator, the piston can not reach the target position quickly, resulting in overshoot. When the depth is shallow, the time that P dominates is relatively short, and the D link works when the piston has not reached the minimum position, so that the piston can reach the target position in a relatively short time without overshooting.

Round 2

Reviewer 2 Report

Comments have been considered, flaws have been corrected. I recommend accepting the article in this version.